# Continuous Positive Airway Pressure Reduces Plasma Neurochemical Levels in Patients with OSA: A Pilot Study

**DOI:** 10.3390/life13030613

**Published:** 2023-02-22

**Authors:** Wen-Te Liu, Huei-Tyng Huang, Hsin-Yi Hung, Shang-Yang Lin, Wen-Hua Hsu, Fang-Yu Lee, Yi-Chun Kuan, Yin-Tzu Lin, Chia-Rung Hsu, Marc Stettler, Chien-Ming Yang, Jieni Wang, Ping-Jung Duh, Kang-Yun Lee, Dean Wu, Hsin-Chien Lee, Jiunn-Horng Kang, Szu-Szu Lee, Hsiu-Jui Wong, Cheng-Yu Tsai, Arnab Majumdar

**Affiliations:** 1School of Respiratory Therapy, College of Medicine, Taipei Medical University, Taipei 110301, Taiwan; 2Sleep Center, Shuang Ho Hospital, Taipei Medical University, New Taipei City 235041, Taiwan; 3Division of Pulmonary Medicine, Department of Internal Medicine, Shuang Ho Hospital, Taipei Medical University, New Taipei City 235041, Taiwan; 4Research Center of Artificial Intelligence in Medicine, Taipei Medical University, Taipei 110301, Taiwan; 5Department of Medical Physics and Biomedical Engineering, University College London, London WC1E 6BT, UK; 6Solomon H. Snyder Department of Neuroscience, Johns Hopkins University School of Medicine, Baltimore, MD 21205, USA; 7Department of Neurology, Shuang Ho Hospital, Taipei Medical University, New Taipei City 235041, Taiwan; 8Department of Neurology, School of Medicine, College of Medicine, Taipei Medical University, Taipei 110301, Taiwan; 9Taipei Neuroscience Institute, Taipei Medical University, Taipei 110301, Taiwan; 10Dementia Center, Shuang Ho Hospital, Taipei Medical University, New Taipei City 235041, Taiwan; 11Department of Medical Imaging and Intervention, Chang Gung Memorial Hospital at Linkou, Taoyuan 333423, Taiwan; 12Department of Civil and Environmental Engineering, Imperial College London, London SW7 2AZ, UK; 13Department of Psychology, National Chengchi University, Taipei 11605, Taiwan; 14Chemical Engineering and Biotechnology, University of Cambridge, Cambridge CB2 1TN, UK; 15Cognitive Neuroscience, Division of Psychology and Language Science, University College London, London WC1E 6BT, UK; 16Department of Psychiatry, Taipei Medical University Hospital, Taipei 110301, Taiwan; 17Department of Physical Medicine and Rehabilitation, Taipei Medical University Hospital, New Taipei City 235041, Taiwan; 18Graduate Institute of Nanomedicine and Medical Engineering, College of Biomedical Engineering, Taipei Medical University, Taipei 110301, Taiwan

**Keywords:** obstructive sleep apnea (OSA), neurodegenerative diseases, continuous positive airway pressure (CPAP), total tau (T-Tau), amyloid beta-peptide 42 (Aβ_42_)

## Abstract

Obstructive sleep apnea (OSA) is a risk factor for neurodegenerative diseases. This study determined whether continuous positive airway pressure (CPAP), which can alleviate OSA symptoms, can reduce neurochemical biomarker levels. Thirty patients with OSA and normal cognitive function were recruited and divided into the control (*n* = 10) and CPAP (*n* = 20) groups. Next, we examined their in-lab sleep data (polysomnography and CPAP titration), sleep-related questionnaire outcomes, and neurochemical biomarker levels at baseline and the 3-month follow-up. The paired t-test and Wilcoxon signed-rank test were used to examine changes. Analysis of covariance (ANCOVA) was performed to increase the robustness of outcomes. The Epworth Sleepiness Scale and Pittsburgh Sleep Quality Index scores were significantly decreased in the CPAP group. The mean levels of total tau (T-Tau), amyloid-beta-42 (Aβ_42_), and the product of the two (Aβ_42_ × T-Tau) increased considerably in the control group (ΔT-Tau: 2.31 pg/mL; ΔAβ_42_: 0.58 pg/mL; ΔAβ_42_ × T-Tau: 48.73 pg^2^/mL^2^), whereas the mean levels of T-Tau and the product of T-Tau and Aβ_42_ decreased considerably in the CPAP group (ΔT-Tau: −2.22 pg/mL; ΔAβ_42_ × T-Tau: −44.35 pg^2^/mL^2^). The results of ANCOVA with adjustment for age, sex, body mass index, baseline measurements, and apnea–hypopnea index demonstrated significant differences in neurochemical biomarker levels between the CPAP and control groups. The findings indicate that CPAP may reduce neurochemical biomarker levels by alleviating OSA symptoms.

## 1. Introduction

Obstructive sleep apnea (OSA) is characterized by disordered breathing triggered by upper airway obstruction [1]. In the United States, the prevalence of OSA is approximately 17% and 34% in men and women aged between 30 and 70 years, respectively, and more than 40% and 25% in men and women aged above 50 years, respectively [2]. Although most patients with OSA experience mild symptoms, OSA is a risk factor for various comorbidities, such as cardiovascular disease, metabolic disorder, and neurocognitive impairment [2,3]. A review study summarized the outcomes of clinical and animal studies and indicated that OSA may causally affect the incidence of Alzheimer’s disease (AD) [4]. Moreover, a study reported a relationship between OSA and cognitive function decline [5].

Typical manifestations of OSA include snoring, intermittent hypoxia, arousal, and sleep fragmentation, and all of these may affect neurocognitive functioning. For example, apnea or hypopnea interrupts the oxygen supply in circulation, resulting in hypercapnia [6]. Repetitive hypoxia and reoxygenation occur along with hypercapnia, resulting in reactive oxygen species generation, followed by oxidase overactivation and regional oxidative stress elevation [7]. Furthermore, recurrent hypoxia and elevated oxidative stress have been reported to be associated with a neuroinflammatory reaction [8]. OSA may lead to cognitive dysfunction by damaging the hippocampus, cerebral cortex, or other brain regions due to chronic intermittent hypoxia and neuroinflammation [9]. Previous neurological studies have reported similar outcomes regarding the relationship between OSA and neurocognitive function. For instance, studies have reported that acute sleep deprivation increased the level of Tau protein (T-tau), which is associated with neurodegeneration, and chronic sleep deprivation accelerated T-tau deposition in neural networks [10,11]. Sleep arousal caused by OSA has been reported to increase the deposition of amyloid beta (Aβ), a neurochemical biomarker of neurodegenerative diseases [12]. Another relevant study analyzed the outcomes of polysomnography (PSG) and reported that the apnea–hypopnea index (AHI) was significantly correlated with the T-Tau level (r = 0.34, *p* < 0.01) [13]. All these aforementioned findings indicate that because OSA is associated with the development of neurodegenerative diseases, appropriate OSA treatment may reduce the risks of neurodegenerative diseases.

Continuous positive airway pressure (CPAP) treatment is the clinical standard noninvasive therapy for OSA. CPAP treatment holds the collapsed airway by maintaining positive pressure during respiration, thus alleviating hypoxia risk and improving sleep quality [14]. CPAP intervention can alleviate cognitive impairment caused by OSA. For example, CPAP may delay the onset age of mild cognitive impairment or AD by approximately 10 years (72 vs. 82 years; *p* < 0.01) [15]. In patients with AD, CPAP treatment delayed cognitive decline over a 3-year follow-up period (*p* < 0.05) [16]. Compared with no use of CPAP, the long-term use of CPAP improved sleep quality and delayed cognitive decline [17]. However, the relationship between CPAP intervention and alterations in neurochemical biomarker levels remains unclear.

This pilot study investigated whether CPAP intervention can improve blood biomarker levels (objective) and sleep questionnaire responses (subjective). We hypothesize that the use of CPAP would reduce the concentrations of neurochemical biomarkers and thus indirectly reduce the risks of AD and other neurodegenerative diseases in patients with OSA. This study enrolled patients with OSA and divided them into two groups on the basis of their therapy type: control and CPAP groups. Next, we examined their PSG and CPAP titration study results as well as their neurochemical biomarker levels and questionnaire outcomes at two time points, namely baseline and after 3 months. A comparison of the parameters between the groups can help determine whether CPAP intervention can alleviate OSA symptoms and thus reduce the risks of neurodegenerative diseases. 

## 2. Materials and Methods

### 2.1. Participants and Study Flow

From the outpatient department of Taipei Medical University, Shuang Ho Hospital (New Taipei City, Taiwan), we recruited patients who were suspected of having OSA and underwent PSG at the sleep center between June and October 2018 and between July 2020 and March 2022. The inclusion criteria were as follows: (1) being aged between 18 and 80 years; (2) not using psychotropic or hypnotic medications in the past 6 months; (3) not undergoing otorhinolaryngological surgery for OSA treatment; (4) not having any cognitive symptoms or a diagnosis of neurological, mental, or psychological disorders (e.g., AD, dementia, epilepsy, and Parkinson’s disease), and (5) not having chronic comorbidities (e.g., cardiovascular diseases, stroke, and metabolic syndrome). The flowchart of patient enrolment is illustrated in Figure 1. All eligible participants were administered the Mini-Mental State Examination (MMSE) [18] to examine their cognitive function as well as overnight PSG to determine their OSA severity. On the following day, at a fixed time (6:30 a.m. when PSG ended), blood samples were collected from all patients to measure neurochemical biomarker levels. Subsequently, patients were administered the Pittsburgh Sleep Quality Index (PSQI) [19] and the Epworth Sleepiness Scale (ESS) [20]. At the next outpatient follow-up (in a week), their PSG results were interpreted, and the OSA treatment was confirmed by attending physicians. Patients made all therapy-related decisions, and medical professionals only provided sufficient relevant information. Next, patients were assigned to the CPAP or control group on the basis of their treatment choice and were followed up for 3 months. At the 3-month follow-up, we examined patients’ neurochemical biomarker levels again and administered the sleep-related questionnaires. We compared the findings between the CPAP and control groups to investigate the effect of CPAP.

Patients were first instructed to conduct Mini-Mental State Examination to exclude patients with cognitive impairment symptoms. Next, all the eligible patients were requested to undergo polysomnography and blood sampling (the following day at a fixed time) and respond to all the questionnaires. After the 3-month follow-up with or without continuous positive airway pressure (CPAP), the second-time data, including blood and questionnaire outcomes, were obtained. All the derived data were divided and compared based on their therapy strategy (i.e., control or CPAP groups). 

### 2.2. PSG Parameters

PSG was conducted using two systems at our sleep center, namely ResMed Embla N7000 (ResMed, San Diego, CA, USA) and an Embla MPR (ResMed Global Supplier Alliance, Sydney, Australia), and the results were scored using RemLogic software (version 3.41; Embla Systems, Thornton, CO, USA). Certified PSG technologists, who underwent regular interscoring training, scored all PSG outcomes in accordance with the 2017 guidelines of the American Association of Sleep Medicine [21]. All scoring outcomes were independently reviewed by another technologist to reduce scoring bias. Inconsistent scoring outcomes were extracted and discussed to determine the final scoring results. Details regarding sleep architecture and the sleep disorder index were obtained. Sleep efficiency, duration in the awake state, and sleep stage distribution (i.e., rapid eye movement [REM] and non-REM [NREM]) were determined. To determine OSA severity, the values of the AHI and oxygen desaturation (ODI-3%) and the data related to snoring and arousal were collected. OSA severity was categorized as mild (AHI = 5 to 15 events/h), moderate (AHI = 15 to 30 events/h), or severe (AHI ≥ 30 events/h) [22].

### 2.3. Determination of Neurochemical Biomarker Levels in Plasma

A hematological biomarker examination was conducted using MagQu Co., Ltd. (New Taipei City, Taiwan) with IMR technology. In EDTA-containing tubes, 16 mL of whole blood was collected. The samples were centrifuged at 2500× *g* for 15 min at room temperature within 15 min of drawing blood. Plasma was withdrawn and stored at −80 °C [23]. Frozen plasma aliquots were shipped on dry ice to MagQu Co., Ltd. for IMR assay. The principle and procedure of IMR are described in a previous study [24]. The reagent of IMR is a solution containing magnetic nanoparticles, which are coated with a bioprobe (e.g., antibodies) and oscillate under external magnetic fields. Once these magnetic nanoparticles associate with target antigens, they aggregate and increase in size. Thus, their response to external magnetic fields is weaker than that of original individual ones. In other words, the more the amount of target antigens in the reagent is, the more the aggregation of magnetic nanoparticles occurs; this results in a larger reduction in magnetic flux for reagents. The reduction of the magnetic property of the reagent indicates the amount of the target antigen in the reagent. The concentration of the target protein is determined based on the reduction in the magnetic field. The present study determined the concentrations of AB_42_ and T-tau protein in plasma. Subsequently, because other studies have indicated that the ratios and products of Aβ_42_ and T-tau may serve as indicators for screening the risk of AD [25,26], the present study further computed these values; that is, this study determined the products and ratios of T-Tau and Aβ_42_ (Aβ_42_ × T-Tau and Aβ_42_/T-Tau, respectively).

### 2.4. CPAP Treatment

For patients who received CPAP therapy, a CPAP titration study was performed to determine the optimal pressure point for employing CPAP. All the procedures were similar to PSG, and a difference existed with respect to the use of CPAP during examination. PSG technologists increased pressure when respiratory events (apnea, hypopnea, and snoring) occurred and determined optimal titration pressure, which reduced the AHI to <5 for at least 15 min (including REM sleeping in the supine position and without spontaneous arousal or awakening). After determining the optimal pressure for each patient, CPAP was administered (APEX-iCH-Auto-2, New Taipei City, Taiwan) for 3 months. All participants were followed up weekly or biweekly to alleviate any relevant risks of using CPAP. All patients were asked to wear a CPAP mask for at least 4 h per night on 5 days per week. The participants who agreed to wear a CPAP mask were visited weekly or biweekly to ensure that they were using the device correctly and to verify their usage time. Those who declined to receive CPAP therapy (control group) received health hygiene education relevant to body weight control and pharmacological support (nasal spray) under the advice of physicians.

### 2.5. Statistics

All statistical analyses were performed using SPSS version 20 (SPSS, Chicago, IL, USA). The Shapiro–Wilk test was first conducted to examine normality. To examine the mean difference in demographic characteristics and PSG outcomes, Student’s t test (continuous variables with a normal distribution), Mann–Whitney U test (continuous variables without a normal distribution), and chi-square test (categorical variable) were employed. Next, to compare changes between baseline and 3-month follow-up measurements in both the control and CPAP groups, the paired t-test was applied for the normally distributed data, whereas the Wilcoxon signed-rank test was used for non-normally distributed data. Analysis of covariance (ANCOVA) was performed to compare alterations in neurochemical biomarker levels between the groups. The equality of slopes of derived data was determined to confirm that the assumptions were met. Neurochemical biomarker levels of the two groups determined at the 3-month follow-up were compared using ANCOVA, with adjustment for age, body mass index (BMI), sex, AHI, and baseline neurochemical biomarker levels as covariates, and the post hoc Bonferroni-adjusted t-contrasts were computed for comparing the main effects. 

## 3. Results

### 3.1. Demographic Characteristics

Table 1 presents the demographic characteristics of the recruited patients. Thirty adults with OSA and typical cognitive function were recruited. Of them, 20 and 10 patients were included in the CPAP and control groups, respectively. No significant differences in age, sex, body mass index, neck circumference, waist circumference, and MMSE scores were observed between the groups. Regarding the distribution of sleep stages, the percentage of the NREM stage was higher in the control group than in the CPAP group (77.86% ± 6.14% vs. 68.03% ± 12.34%, *p* < 0.05). The values of the AHI, ODI-3%, snoring index, and arousal index and the distribution of OSA severity did not significantly differ between the groups. 

### 3.2. Details of CPAP Titration

Table 2 presents the outcomes of CPAP titration for the CPAP group. The mean of optimal CPAP pressure setting was 8.95 ± 3.65 cmH_2_O. During CPAP treatment, sleep efficiency increased from 74.78% ± 16.17% to 79.78% ± 11.96%, the AHI decreased from 47.08 ± 20.38 to 5.39 ± 4.19 events/h, the ODI-3% decreased from 44.62 ± 20.67 to 6.22 ± 5.4 events/h, and the mean arousal index decreased from 27.42 ± 11.51 to 9.02 ± 5.87 events/h. Regarding CPAP mask usage time, as mentioned, all patients were instructed to wear a CPAP mask for at least 4 h per night on 5 days per week. Of the participants, 70% wore the CPAP mask for 4–6 h per night, and 30% wore it for 6–8 h per night.

### 3.3. Alterations in the Neurochemical Biomarker Levels

Table 3 presents the details of neurochemical biomarker levels and questionnaire responses at baseline and the 3-month follow-up for both the groups. The mean ESS and PSQI scores did not significantly differ between baseline and the 3-month follow-up in the control group. However, the CPAP group had significantly lower mean ESS (baseline: 13.25 ± 6.15, 3-month follow-up: 9.5 ± 6.13, *p* < 0.05) and PSQI (baseline: 10.05 ± 4.05, 3-month follow-up: 6.85 ± 3.56, *p* < 0.01) scores, respectively. Regarding neurochemical biomarkers, the levels of T-Tau (*p* < 0.05), Aβ_42_ (*p* < 0.05), and the product (Aβ_42_ × T-Tau, *p* < 0.01) were significantly increased at the 3-month follow-up in the control group. By contrast, the levels of T-Tau (*p* < 0.05) and the product (Aβ_42_ × T-Tau, *p* < 0.05) were significantly decreased, and the ratio of Aβ_42_ to T-Tau was significantly increased (*p* < 0.01) in the CPAP group. 

### 3.4. Effect on CPAP on Neurochemical Biomarker Levels

In this study, ANCOVA was used to investigate whether alterations in neurochemical biomarker levels at baseline and the 3-month follow-up differed between the control and CPAP groups, and the results are displayed in Figure 2. The mean change in the T-Tau level (∆T-Tau) between baseline and the 3-month follow-up significantly differed between the CPAP group (mean: −2.22 ± 3.85 pg/mL) and the control group (mean: 2.31 ± 2.16 pg/mL). Similarly, changes in the levels of the product of T-Tau and Aβ_42_ (∆Aβ_42_ × T-Tau) significantly differed between the CPAP (mean: −44.35 ± 88.92 pg/mL) and control (mean: 48.73 ± 34.02 pg/mL) groups. Furthermore, changes in the ratio of Aβ_42_ to T-Tau (∆Aβ_42_/T-Tau) significantly differed between the CPAP (mean: 0.05 ± 0.07 pg/mL) and control (mean: −0.06 ± 0.09 pg/mL) groups.

The alterations in the neurochemical biomarker levels were determined by calculating differences in the measurements of baseline and after 3 months, including (A) alterations of T-Tau, (B) alterations of Aβ_42_, (C) alterations of T-Tau X Aβ_42_, and (D) alterations of T-Tau/Aβ_42_. The derived mean and quantile values were demonstrated. The analysis of covariance (ANCOVA) was applied to determine the statistical difference by adjusting the age, sex, body mass index and apnea–hypopnea index to investigate the effect on biomarker level alterations of employing CPAP. 

## 4. Discussion

OSA is associated with elevated neurochemical biomarker levels and can thus increase the risk of neurodegenerative diseases. Thus, treating OSA (e.g., by employing CPAP) may reduce neurochemical biomarker levels. To investigate this possibility, in this pilot study, we recruited patients with OSA (diagnosed on the basis of PSG results), measured their plasma neurochemical biomarker levels, and provided them with therapy options (CPAP or health hygiene education). The findings revealed that at the 3-month follow-up, the CPAP group had significantly decreased mean levels of T-Tau and the product of T-Tau and Aβ_42_, whereas the control group had significantly increased mean levels of T-Tau, Aβ_42_, and their product. Furthermore, to compare the effect of CPAP intervention between the groups, we performed ANCOVA with adjustment for age, sex, BMI, baseline neurochemical biomarker levels, and the AHI. Changes in neurochemical biomarker levels (between baseline and the 3-month follow-up) significantly differed between the CPAP and control groups.

The present study compared baseline and 3-month follow-up measurements in the control group, and the results revealed increased mean levels. These outcomes are in line with those of previous studies examining the association between untreated OSA and elevated neurochemical biomarker levels [27]. In other words, the clinical symptoms of OSA, including sleep-disordered breathing, nighttime hypoxia, sleep arousal, and sleep fragmentation, may be risk factors for neurochemical biomarker formulation and accumulation [28]. A related study reported a significant association between a high mean AHI and increased neurochemical biomarker levels in patients with OSA without dementia [29]. Another study analyzed PSG parameters and neurochemical biomarker levels and suggested that the elevated AHI, ODI-3%, and arousal indexes were significantly associated with the risk of AD [30]. OSA acts in synergism with neurochemical biomarkers (i.e., Aβ_42_ and T-tau), and patients with mild cognitive impairment having untreated OSA may have shorter progression time to AD compared with those with only mild cognitive impairment [31]. Taken together, the findings indicate that untreated OSA, which can continue to cause sleep-disordered breathing and other manifestations (e.g., arousal response and oxygen desaturation), may interrupt the neurochemical biomarker removal process and, thus, increase the risk of neurodegenerative diseases.

In this study, the CPAP group exhibited significantly decreased mean levels of T-tau and the Aβ_42_ × T-Tau product. Moreover, the CPAP group exhibited improved ESS and PSQI scores at the 3-month follow-up. Similarly, the results of ANCOVA with adjustment for age, sex, BMI, baseline neurochemical biomarker levels, and the AHI revealed that the CPAP group had decreased neurochemical biomarker levels than did the control group. CPAP therapy is a noninvasive option for patients with OSA in clinical practice and is effective in multiple aspects. For example, CPAP prevents upper airway occlusion during sleep and enables patients to have an entire night of uninterrupted sleep [32]. CPAP considerably improves sleep quality [33]. Another study evaluated treatment responses by calculating various questionnaire outcomes (i.e., mood, energy or fatigue, functional status, and general health), and the results suggested that the odds of experiencing CPAP treatment responses were approximately three times compared with conservative treatment responses [34]. CPAP users had lower odds of incident diagnoses of AD (odds ratio [OR] = 0.78, 95% confidence interval [95% CI] = 0.69 to 0.89) and mild cognitive impairment (OR = 0.82, 95% CI = 0.66 to 1.02) [35]. A related systematic review indicated that receiving CPAP intervention for OSA treatment reduced the risk of AD [36]. Collectively, the results indicate that employing CPAP intervention to alleviate OSA symptoms and improve sleep quality can indirectly prevent neurodegenerative diseases by reducing neurochemical biomarker levels. 

The present study had some strengths. Although previous researchers have indicated that CPAP therapy partially reverses neurochemical biomarker levels [37], to the best of our knowledge, the present study was the first to investigate the effects of CPAP therapy on both plasma neurochemical biomarker levels (objective measure) and questionnaire responses (subjective measure). In addition, other observational studies that have employed CPAP therapy have observed changes only in levels of Aβ_42_ [38,39], whereas the present study identified changes in both Aβ_42_ and T-Tau protein levels. Furthermore, although other studies have obtained similar findings indicating that CPAP therapy may reduce the levels of Aβ_42_ and T-Tau in patients with OSA combined with mild cognitive impairment [40], the present study enrolled only patients with OSA who had no cognitive problems and, therefore, was able to provide evidence that CPAP interventions can benefit patients with OSA alongside typical cognitive function and provide early prevention of cognitive impairment. The present study further observed that untreated OSA may negatively affect neurochemical biomarker levels and sleep quality. Furthermore, in contrast to other studies that have investigated the effects of OSA on the incidence of neurodegenerative diseases [41,42], the present study investigated the benefits of CPAP therapy and provided evidence supported by data regarding neurochemical biomarker levels rather than that supported by only data regarding disease prevalence. Because the benefits of CPAP intervention have not been thoroughly investigated and because existing evidence of these benefits remains preliminary, the present results offer crucial evidence indicating that CPAP intervention can not only alleviate OSA manifestations but also prevent cognitive problems by reducing neurochemical biomarker levels. The results further indicate that untreated OSA can interfere with sleep efficiency, aggravate OSA manifestations, and affect neurochemical biomarker levels.

In terms of limitations that should be considered, this study did not examine the effects of genetic factors on neurochemical biomarker levels. However, previous related studies have reported associations between neurodegenerative diseases and some specific genes (e.g., FABP3 and ApoE4) [43]. Next, although neurochemical biomarkers in both the cerebrospinal fluid and plasma are regarded as reliable tools for screening the risk of some neurodegenerative diseases, differences might exist between the two measurements [44]. We measured the neurochemical biomarker levels in plasma only; we did not measure them in cerebrospinal fluid. A growing body of evidence suggests that the presence of neurochemical biomarkers in cerebrospinal fluid is an indicator of a continuum of cognitive disorders and, particularly, an indicator of AD pathology [45,46]. However, practical clinical obstacles—such as invasiveness, time, and cost—limited the feasibility of performing lumbar puncture to obtain cerebrospinal fluid in the present study [47]. Therefore, this study employed an alternative approach that was rapid, minimally invasive, and low in cost to determine the levels of plasma-based biomarkers for the analyses. Nevertheless, neurochemical biomarker levels in cerebrospinal fluid may be more directly indicative of cognitive function or neurodegenerative diseases. In addition, the present study featured the limitation of a small sample size, and all the patients were enrolled from a single sleep center in northern Taiwan. The low enrollment may have been the result of CPAP adherence limitations (CPAP therapy often involves discomfort and nasal congestion) and of potential participants being informed before enrollment that they would be required to use the CPAP device for a certain period to ensure participant adherence. Nevertheless, because of these factors, although this study recruited a small number of patients, all the recruited patients met the criterion related to CPAP use time. However, this limitation may affect the generalizability of our outcomes for different populations. Although this pilot study excluded patients with comorbidities or regular medication usage, lifestyle habits (tobacco and alcohol use) may affect OSA severity or neurochemical biomarker levels [48]. Thus, to enhance the robustness of the derived statistical results, these lifestyle factors should be further examined, and a multicenter study including more participants or patients of various ethnicities should be conducted. 

## 5. Conclusions

The Taiwanese patients recruited to this study were administered PSG and received OSA treatment for 3 months (CPAP group) or health hygiene education (control group). In addition, we measured two neurochemical biomarker levels and questionnaire responses at baseline and the 3-month follow-up. The CPAP group had significantly lower mean ESS and PSQI scores at the 3-month follow-up than at baseline. In the control group, the mean levels of T-Tau, Aβ_42_, and their product significantly increased at the 3-month follow-up than at baseline. The CPAP group exhibited significantly decreased levels of T-Tau and T-Tau × Aβ_42_ at the 3-month follow-up. This study used ANCOVA to determine whether CPAP can reduce neurochemical biomarker levels. After adjustment for age, sex, BMI, AHI, and baseline neurochemical biomarker levels, alterations (between the 3-month follow-up and baseline) in the CPAP group were significantly different than those in the control group, including T-Tau, the computed product (Aβ_42_ × T-Tau), and the ratio of Aβ_42_ to T-Tau. Regarding the clinical significance of this pilot study, the major findings of this study indicate that the use of CPAP devices may reduce neurochemical biomarker levels and improve sleep quality by alleviating OSA symptoms. Furthermore, the findings indicate that for patients with OSA who do not undergo treatment, OSA is likely to affect sleep quality, and the manifestations of OSA are likely to intensify and increase neurochemical biomarker levels. However, a longitudinal study with a larger population data set should be conducted to verify the causal relationships among these factors.

## Figures and Tables

**Figure 1 life-13-00613-f001:**
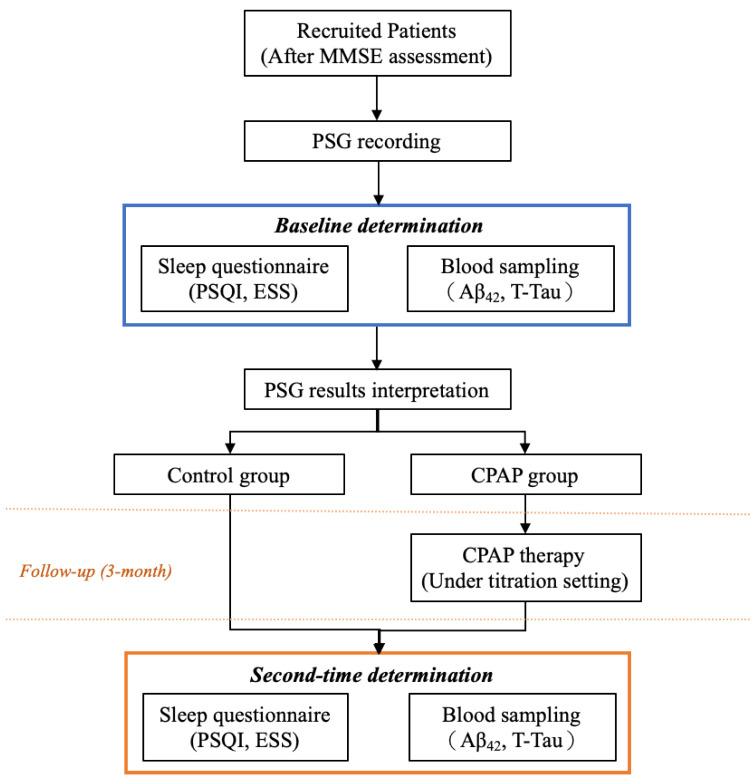
Flowchart of clinical assessment, treatment, and follow-up details of this study. Abbreviations: MMSE: Mini-Mental State Examination; PSG: polysomnography; PSQI: Pittsburgh Sleep Quality Index; ESS: Epworth Sleepiness Scale; T-Tau: total tau; Aβ42: amyloid-beta-42; CPAP: continuous positive airway pressure.

**Figure 2 life-13-00613-f002:**
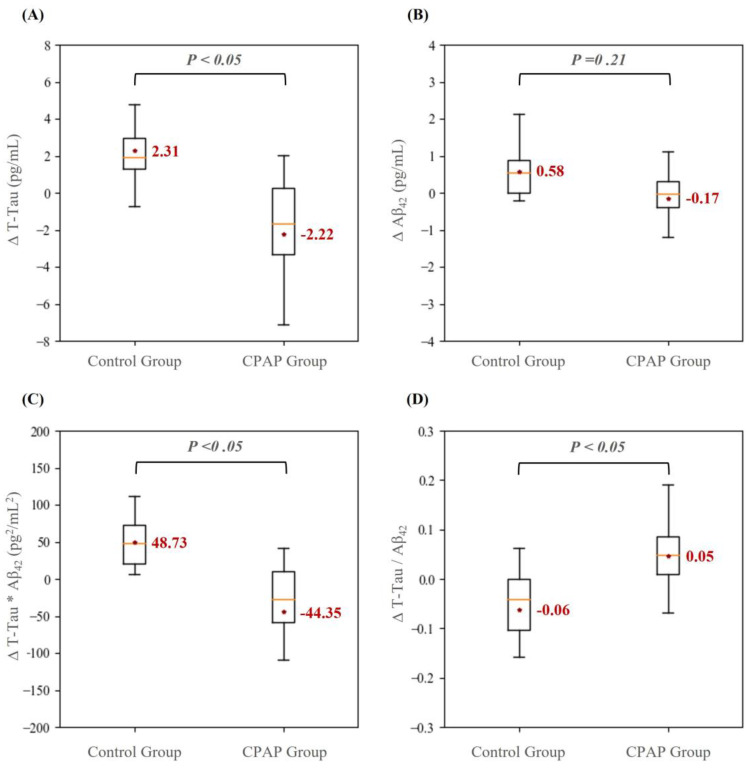
Comparison of the alteration values (Δ) at two time points for levels of neurochemical biomarkers—namely (**A**) T-Tau protein, (**B**) Aβ_42_, (**C**) Aβ_42_ × T-Tau, and (**D**) Aβ_42_/T-Tau— between the control and continuous positive airway pressure (CPAP) groups.

**Table 1 life-13-00613-t001:** Demographic characteristics between the control and continuous positive airway pressure (CPAP) group.

Categorical Variables	Control Group (*n* = 10)	CPAP Group (*n* = 20)	*p*
Age (year) ^a^	50.4 ± 11.84	53.85 ± 10.53	0.43
Sex (male/female) ^b^	9/1	16/4	0.48
BMI (kg/m^2^) ^c^	27.78 ± 3.86	30.42 ± 4.41	0.12
Neck circumference (cm) ^c^	39.7 ± 3.13	40.4 ± 4.0	0.63
Waist circumference (cm) ^c^	98.5 ± 10.3	100.32 ± 12.15	0.69
MMSE (score) ^a^	29.2 ± 1.03	28.5 ± 1.93	0.43
Sleep architecture			
Sleep efficiency (%) ^a^	81.62 ± 10.36	74.78 ± 16.17	0.23
Wake (% of SPT) ^a^	12.48 ± 8.13	20.29 ± 14.7	0.17
NREM (% of SPT) ^c^	77.86 ± 6.14	68.03 ± 12.34	<0.05
REM (% of SPT) ^c^	9.64 ± 5.75	11.68 ± 5.73	0.37
WASO (min) ^c^	42.58 ± 27.79	67.65 ± 43.63	0.11
Sleep disorder index (events/h)			
ODI-3% ^c^	41.24 ± 24.14	44.62 ± 20.67	0.69
AHI ^c^	42.02 ± 22.42	47.08 ± 20.38	0.54
Snoring index ^c^	299.8 ± 292.06	374.7 ± 154.82	0.36
Arousal index ^c^	29.19 ± 18.0	27.42 ± 11.51	0.75
OSA severity ^b^			0.05
Normal, *n* (%)	-	-	
Mild, *n* (%)	2 (20%)	-	
Moderate, *n* (%)	-	4 (20%)	
Severe, *n* (%)	8 (80%)	16 (80%)	

Abbreviations: BMI: body mass index; MMSE: Mini-Mental State Examination; SPT: sleep period of time; NREM: Nonrapid eye movement; REM: Rapid eye movement; WASO: wake time after sleep onset; ODI-3%*:* oxygen desaturation index ≥3%; AHI: apnea–hypopnea index; OSA: obstructive sleep apnea. Data are expressed as the mean ± standard deviation. Differences between groups were assessed using the ^a^ Mann–Whitney U test, ^b^ chi-squared test, and ^c^ Student’s *t*-test.

**Table 2 life-13-00613-t002:** Sleep parameters and ventilator setting details in continuous positive airway pressure (CPAP) titration.

Categorical Variables.	CPAP Group (*n* = 20)
CPAP setting (cmH_2_O)	
Optimal CPAP level	8.95 ± 3.65
Pressure 95th centile	9.06 ± 3.6
Maximum pressure	10.58 ± 4.65
Sleep architecture	
Sleep efficiency (%)	79.78 ± 11.96
Wake (% of SPT)	16.6 ± 10.82
NREM (% of SPT)	67.73 ± 8.25
REM (% of SPT)	15.66 ± 6.03
WASO (min)	58.76 ± 36.52
Sleep disorder index (events/h)	
ODI-3%	6.22 ± 5.4
AHI	5.39 ± 4.19
Snoring index	73.4 ± 113.06
Arousal index	9.02 ± 5.87
CPAP usage time (*n*, %)	
4–6 h	14 (70%)
6–8 h	6 (30%)

Abbreviations: SPT: sleep period of time; NREM: Non-rapid eye movement; REM: Rapid eye movement; WASO: wake time after sleep onset; ODI-3%*:* oxygen desaturation index ≥3%; AHI: apnea–hypopnea index; Data are expressed as the mean ± standard deviation.

**Table 3 life-13-00613-t003:** Comparison of changes in the sleep-related questionnaire scores and neurochemical biomarker levels between the two time points (baseline and after 3 months) in the control and continuous positive airway pressure (CPAP) groups.

Categorical Variables	Control Group (*n* = 10)			CPAP Group (*n* = 20)		
	Baseline	After 3-Month	*p*	Baseline	After 3-Month	*p*
Questionnaire						
ESS (score) ^a^	10.5 ± 7.07	9.2 ± 6.7	0.32	13.25 ± 6.15	9.5 ± 6.13	<0.05
PSQI (score) ^a^	8.6 ± 3.57	7.7 ± 2.98	0.56	10.05 ± 4.05	6.85 ± 3.56	<0.01
Neurochemical biomarker						
T-Tau (pg/mL) ^b^	20.17 ± 3.48	22.48 ± 2.78	<0.05	24.69 ± 5.26	22.47 ± 2.38	<0.05
Aβ_42_ (pg/mL) ^b^	15.84 ± 1.03	16.43 ± 0.64	<0.05	16.53 ± 1.06	16.36 ± 0.43	0.73
Aβ_42_ × T-Tau (pg^2^/mL^2^) ^b^	321.21 ± 66.36	369.94 ± 51.73	<0.01	412.56 ± 117.73	368.21 ± 46.21	<0.05
Aβ_42_/T-Tau ^a^	0.8 ± 0.12	0.74 ± 0.09	0.07	0.69 ± 0.09	0.73 ± 0.07	<0.01

Abbreviation: ESS: Epworth Sleepiness Scale; PSQI: Pittsburgh Sleep Quality Index; T-Tau: total tau; Aβ_42_: amyloid beta 42. Data are expressed as the mean ± standard deviation. Differences between mean level at baseline and 3-month follow-up were assessed using the ^a^ paired *t*-test and ^b^ Wilcoxon signed-rank test.

## Data Availability

All the data of this study were collected at the Sleep Center of Taipei Medical University–Shuang Ho Hospital (New Taipei City, Taiwan) between May 2019 and December 2021. Because our data set contains personal information, it is not available in the supplement file. For access to the data set or relevant documents, please contact the corresponding author.

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
