# Peer review of "Continuous Positive Airway Pressure Reduces Plasma Neurochemical Levels in Patients with OSA: A Pilot Study"

_life, 2023, doi:10.3390/life13030613_

Round 1

Reviewer 1 Report

First, I would like to compliment the authors for this investigation work.

Abstract: I would recommend including numeric data on the changes in the neurochemical biomarker levels. I find it adequate to include only the conceptual explanation in the simple summary section. However, the abstract section must include the main results with numeric values. 

Section 2.4. Patients were requested to use CPAP for 4 h - 5 days. Was the adherence and time of use registered?

Section 3.1. What happened with the 2 remaining patients? 32 people were included. However, only two groups of 10 and 20 are described.

Figure 2. It is a Litle bit confusing; please, explain better what represents a), b), c), and d). The content is similar to Table 3, so it could even be eliminated.

Author Response

We appreciate your valuable suggestions. We have carefully revised our manuscript per your feedback.

Reviewer 2 Report

The manuscript is clear. The study is rigorous, original, with interesting conclusions.A larger sample is needed.

Author Response

(The authors gave the same response as above.)

Reviewer 3 Report

Dear Author (s)

1. There are recent meta-analyses about this subject. What is novelty?

2. Why did you measure plasma level, not CSF?

3. Compared to other studies, you have lower cases. Why?

4. Some references in the introduction are very old. Please change them with ones newer.

5. You missed some recent meta-analyses and reviews for citations.

6. Please describe :Aβ42 × T-Tau or Aβ42 / T-Tau. What are they showing?

7. What is clicinal significance?

Author Response

(The authors gave the same response as above.)

Round 2

Reviewer 3 Report

Dear Author(s)

The manuscript is acceptable.